# Investigation of the associations between physical activity, self-regulation and educational outcomes in childhood

Fotini Vasilopoulos[1,2]*, Michelle R. Ellefson[1]

**1** Faculty of Education, University of Cambridge, Cambridge, United Kingdom, **2** Department of Psychological Sciences, University of London, London, United Kingdom

* fv263@alumni.cam.ac.uk

**Data Availability Statement:** The data that support the findings of this study are available from the UK Data Service repository. Restrictions apply to the availability of these data, which were used under licence for this study. The authors of this study are not the owners of this data and had to make a

## Abstract

It is common knowledge that physical activity leads to physiological and psychological benefits. The current study explored the association between physical activity and self-regulation longitudinally and the indirect relationship this may have on academic achievement, using secondary data on primary and secondary school children from the Millennium Cohort Study, a cohort of infants born in 2000–2001 in the United Kingdom. There are two main findings. First, there is a positive link between physical activity and emotional (not behavioural) regulation both concurrently and longitudinally across all three time points, 7-years-old, 11-years-old and 14-years-old. The relationship was negative for emotional regulation and negligible for behavioural regulation when controlling for socioeconomic status. Second, across two time points (due to data availability), physical activity positively predicted academic achievement through emotional regulation for 7-year-olds and behavioural regulation in 11-year-olds. The impact of this relationship was more pronounced when controlling for socioeconomic status. Together these findings indicate that emotional regulation is linked to physical activity in early childhood. Subsequently, emotion regulation predicts academic attainment, suggesting that early interventions might focus on attention rather than behaviour.

## Introduction

Physical activity (PA), particularly the influence it may have on mental processes, has interested researchers for a few decades, with a recent resurgence taking place due to technological advancements in brain imaging technology and an increased focus on health and wellness [1]. Studies using functional magnetic resonance imaging indicate that exercise boosts activation in children's prefrontal cortex (part of the autonomic nervous system) [2] which is associated with self-regulatory behaviours [3]. This area is also affected when someone is not physically fit [4]. The socioeconomic gradient for PA has increased during recent decades: compared to 13% of high-socioeconomic status (SES) children, 34% of low-SES children in the United Kingdom participate in sports less than once a week [5]. The United Kingdom government PA

specific request to access it for the analyses reported here. Data are available https://beta.ukdataservice.ac.uk/datacatalogue/series/series?id=200003 with the permission of UK Data service repository. Detailed information about the processing of the data used in this manuscript can be found at beta.ukdataservice.ac.uk/datacatalogue/series/series?id=2000031.

**Funding:** The authors received no specific funding for this work.

**Competing interests:** No authors have competing interests.

**Abbreviations:** CFI, Comparative Fit Index; MD, Mahalanobis distance; mg, milligravitational units; RMSEA, Root Mean Square Error of Approximation; SEM, Structural Equation Modelling; SES, Socioeconomic Status; SRMR, Standardized Root Mean Square Residual; TLI, Tucker–Lewis Index.

guidelines recommend that children ages 5–18 years of age should aim for an average of 60 minute of moderate intensity PA per day [6]. Furthermore, Efrat's [7] review of various studies suggests PA and academic outcomes are positively correlated in children from poorer backgrounds. Thus, PA may help disadvantaged children with academic achievement by influencing self-regulation. Understanding the gap by trying to understand what influences self-regulation when considering risk factors could paint a better picture of what contributes to a child optimising outcomes within the context of limited resources.

## Self-regulation

Self-regulation is a skill to control emotions and behaviour dependent on the demands of the situation [8]. Self-regulation can be affected by internal factors including biology and genetics and external factors such as caregiver practices, poverty and school interventions [9]. Developmental studies suggest that children's assets, such as self-regulation, are essential for children's educational success [10]. Self-regulation research includes cognitive and non-cognitive domains that enable one to manage emotions, motivation and cognitive arousal [11] for positive adjustment [12]. Specifically, there are three distinct aspects of self-regulation, including cognitive, emotional and behavioural regulation. Cognitive regulation, commonly known as executive function, is a cognitive skill used to control thought and action to achieve a goal [13]. Emotional regulation is the management of emotional reactions to achieve a goal [14]. Behavioural regulation is the inhibition of behaviour and management of attention to achieve a goal.

Recent work suggests a multi-dimensional approach to self-regulation over a one factor model with three factors being sub-components as each have specific development trajectories [15]. Self-regulation shows steep changes in early childhood and significant improvement throughout childhood and adolescence, suggesting there may be an extended window of plasticity where some aspects of self-regulation may be fostered.

## Physical activity and childhood outcomes

PA is defined as movement in the body that results in an outflow of energy [16]. Recent studies examined the amount of PA [2], intensity [17], content [18], duration [19], environment [20], enjoyment [21], type [22] and the impact on outcomes.

## Physical activity and self-regulation

PA seems to be linked to self-regulation. Systematic and meta-analytical reviews [23] suggest PA-based interventions during childhood and adolescence showed consistent improvement in self-regulation.

The recent body of literature has identified specific parameters of PA and its links to self-regulation, including type, intensity, and environment. Type of PA has been defined in various ways in the literature when investigating the links with outcomes, from a basic view (individual and team sports) to a more complex view (open and closed skilled sports). Open-skilled sports are characterised by their variability and unstable environment (e.g., tennis) whereas closed-skilled sports are considered predictable within a stable environment (e.g., swimming) [22]. The association between type of PA and self-regulation was identified over a two-year period in a longitudinal study conducted by Howard et al. [24] for individual sports and not team sports. Howard and colleagues applied univariate general linear models to a large data set (N = 4,385) with multiple sources to measure self-regulation and found that children aged 4–5 years who played individual sports had significantly better self-regulation two years later than those who did not. However, self-regulation does not seem to be impacted by team sport

participation. Furthermore, there is a correlation between open-skilled sports and self-regulation in 7- to 11-year-old children [2] but not in third-grade children [22].

More intensive PA has also been associated with boosting self-regulation across childhood. This has been evidenced in early childhood after outdoor play [16] and middle childhood when spending more than 60 minutes in moderate to vigorous PA during a school year [17] or participating in various sports [22].

The literature also points to enjoyment of the environment of PA as another factor influencing outcomes [20, 21, 25, 26]. Systematic literature reviews suggest that participating in outdoor PA could have a greater effect on positive mental wellbeing and self-regulation.

## Physical activity and long-term outcomes

PA is important for physiological and psychosocial health outcomes for children [27, 28]. Lower levels of PA are associated with poorer health outcomes such as cardiovascular disease and mortality in middle-aged to older adults [29, 30]. Furthermore, higher amounts of PA in childhood is associated with better bone health, obesity trajectories, cardiovascular health and reduced probability of smoking in early to mid-adulthood [31–33]. This positive association is also seen in better mental health and lower depressive symptoms in the long-term [34]. Educational success has also been linked to PA.

## Physical activity and academic achievement

Different aspects of PA and how they influence academic achievement have been studied extensively. There are a large number of systematic and meta-analytical reviews published in the past ten years with mixed findings linking PA with cognition or academic outcomes in children [35–37]. Although, several studies have demonstrated that participation in more vigorous PA is associated with numeracy and literacy academic achievement in preadolescence and adolescence [35] with the same being true in the younger years [16], others have found that this is only the case for numeracy [38], or neither numeracy or literacy [22].

The type of PA and its influence on academic achievement has not been studied as extensively. Various results suggest that primary school aged children who participate in open-skills sports have better performance in numeracy but not literacy [22, 39]. A systematic review of PA and academic achievement across school aged children found half of the associations were positive and half did not demonstrate a relationship [40]. The results between PA and educational outcomes may be due to elements of PA that were not discussed in the studies identified, such as intensity or duration. Thus, it is possible that the type of PA influences academic achievement in adolescence when intensity and duration is also considered.

No study to date has investigated frequency of PA and academic achievement. One study [41] investigated the link between PA at 7-years-old and cognition at 11-years-old and discovered that frequency of sports club participation was linked with higher cognitive scores. Together, the above research does not confirm a direct link between PA and academic achievement.

## Physical activity, self-regulation and academic achievement

Theoretically, embodied cognition provides support for the relationship between PA, self-regulation and academic achievement. This model suggests that mental processes are supported through the interaction between the body and the external environment, thus creating a link between the mind and the body. Therefore, it could be that self-regulation and academic achievement can be promoted through the body and neural layers related to the motor system [16, 42].

Only one study has analysed PA, self-regulation and outcomes in the early years (pre-kindergarten year) and found a relationship between self-regulation on academic outcomes from PA [16]. To date, no inclusive model has been developed over a long-term timeline with objectively measured PA, which this cohort data set affords [1]. The absence of evidence is not evidence of absence of impact but reflects more on the cost and challenge of life-course studies.

## Current study

The aim of this study is to understand the developmental trajectory for those who are physically active on outcomes during childhood. Consequently, this study is influenced by two lines of inquiry in developmental research. First, to investigate whether PA is associated with self-regulation via the body-mind connection, and that improvement in self-regulation predicts academic outcomes from childhood into early adolescence. Second, the aspects of emotional and behavioural regulation will be studied together to gain a better understanding of the development of self-regulation in relation to PA and its relationship with academic achievement. We have chosen to focus on emotional and behavioural regulation skills as they help with the transition to adulthood so that findings can be integrated with cognitive research to create a developmental picture in a natural context.

Therefore, we examined the a) link between PA and two facets of self-regulation longitudinally from ages 7 to 14 years and b) relationship between PA and academic achievement during primary school education through emotional and behavioural regulation in childhood. Both relationships were also analysed with SES.

## Methods

### Participants

Data for this study came from the *Millennium Cohort Study* (a longitudinal study of children born during 2000–2001 in the United Kingdom [43, 44]). Seven surveys of the *Millennium Cohort Study* have been conducted to date with survey four (7 years; 2008: N = 13,857) being when the first set of PA data was collected and the starting point for this study. All children surveyed at age 7 years were invited to wear an activity monitor (accelerometer). Only children who wore a PA monitor and surveyed in England (n = 4,043, 50% boys, 82% white, 18% ethnic minority) were included in this study to ensure that different schooling systems did not influence the relationships being tested. At age 14 years, a random subsample of 81% of cohort members in England were asked to wear the devices. This dataset was chosen for this study as it included PA data measured objectively, extensive documentation and inclusion of pre-tested measures.

The sample was not designed to be wholly representative of the population to maintain a reasonable representation of social disadvantage as selective attrition is seen in more disadvantaged groups. A stratification process by region was used to ensure that the proportion of children living in disadvantaged areas was over-sampled [45]. In instances of many children in a family, only the data of the first child were included for analysis to avoid clustered data structures [15]. The *Millennium Cohort Study* was approved by the South West, London and Yorkshire Multi-Centre Research Ethics Committees (MREC/01/6/19, MREC/03/2/022, 05/MRE02/46, 07/MRE03/32, 11/YH/0203, 13/LO/1786for sweeps one, two, three, four, five and six respectively). The waves reported were conducted under relevant ethical committee approval from the National Health Service (NHS) Research Ethics Committee system which are appointed by the Strategic Health Authorities in England; parental written, informed consent was obtained of all children.

## Measures

This study incorporates self-regulation as two constructs, emotional and behavioural regulation, building on Edossa et al.'s [15] work, which also used the *Millennium Cohort Study*. Emotional and behavioural aspects of self-regulation were measured at ages 7, 11 and 14 using a combination of teacher–parent reports using a subset of items from the Child Social Behaviour Questionnaire (CSBQ, [46]) and the Strength and Difficulties Questionnaire (SDQ, [47]). Both instruments have been validated for use in England [48, 49]. The reported behaviour instruments used a three-point scale and had similar wording across all time-points. The responses had three categories: "Not True", "True", and "Certainly True". Confirmatory factor analysis performed by Edossa et al. [15] confirmed a good fit on the same subset of items selected as both constructs of self-regulation measures using the same data set in the early years.

**Emotional regulation.** At age 7, parents rated a child's behaviour using the CSBQ for the following four items "The child shows mood swings" (Item 1), "gets over excited" (Item 2), "gets easily frustrated" (Item 3) and "acts impulsively" (Item 4). Teacher's also rated a child's behaviour at age 7 using the SDQ for the following two items "Is restless, overactive, cannot stay still for long" (Item 5) and "Is easily distracted, concentration wanders" (Item 6). One person then rated behaviour, at age 11 (teacher) and 14 (parent), both using the SDQ with the same subset of items at age 7. See the descriptive statistics in the S1 Table. Based on our analyses, the first four items from the CSBQ were excluded because of low factor loading.

**Behavioural regulation.** Like emotional regulation, behavioural regulation was rated at the same time points using the same questionnaires. At age 7 two items, "The child persists in the face of difficult tasks" (Item 1) and "moves to a new activity after finishing a task" (Item 2), a sub-domain of the CSBQ [46]. The items "sees tasks through to the end" (Item 3), "can stop and think before acting" (Item 4) and "often loses temper" (Item 5) were adopted from the SDQ [49]. At age 11 (teacher) and 14 (parent), the SDQ with the same subset of items at age 7 was used. See the descriptive statistics in the S1 Table.

**Physical activity.** PA is represented as one construct in this study to represent a "real life view" of PA and incorporating different parameters of PA that the substantive body of literature has confirmed it effects self-regulation. An index was built using different parameters to fulfil the main aim of this study; understanding the developmental trajectory of PA when disadvantage is at play. An objective and subjective measure was included at each time point to counteract the weaknesses in using subjective measures only such as parent or self-reports [22, 50, 51]. The data set measured parameters which include moderate to vigorous physical activity (MVPA) greater than 60 minutes, enjoyment of PA in different settings and duration of type of PA.

PA intensity (MVPA per day) was measured at two time-points: 7 and 14-years-old by objectively recording activity through a monitor (accelerometer) worn by each participant. At age 7 the Actigraph GT1M uni-axial accelerometer was worn on the right hip for seven consecutive days during waking hours; and at age 14 the GENEActiv tri-axial accelerometer was worn on the wrist of the non-dominant hand for two 24 hour periods, one during a weekday and another during a weekend. The MVPA cut-points used were $\geq 2,240$ and $\geq 3840$ counts per minute for 7-year-olds [52] and Euclidean Norm Minus One $\geq 100$ milligravitational units (mg) for 14-year-olds. Historically, accelerometers cut points were in the form of 'counts', however, recently, there has been a move towards measuring PA using raw acceleration techniques, such as Euclidean Norm Minus One instead of cut-points [53]. Counts are produced by an algorithm whereas raw data is not processed. The cohort study has also adopted this view when collecting PA data at age 14 years. One study [54] investigated acceleration cut points comparing the Actigraph and GENEActive in different populations (children

aged 7–11 years and adults over 18 years) and found that MVPA for adults should be ≥93mg and ≥190mg for children. Guidance for the adolescent age group has not been confirmed. However, one recent study [53] applied acceleration ≥200mg for MVPA to 1,669 adolescences citing the aforementioned research paper. Presumably, the *Millennium Cohort Study* selected 100mg as age 14 years is closer regarding development to adulthood than childhood. Furthermore, The GENEActive and Actigraph accelerometers showed strong agreement in a validation study [55] with intra-class correlation >0.95 [56]. The epoch lengths applied were 15 and 5 seconds at 7 and 14-years-old, respectively.

Subjective measures of enjoyment were also included in the PA variable at both time points. Self-report at age 7 years "how much do you like playing sports and games inside?" (Item 2), see the descriptive statistics in the S1 Table and "how much do you like playing sports and games outside?" (Item 3). Self-reports were provided on a three-point scale ("I like it a lot", "I like it a bit", "I don't like it").

Duration was measured at 14-years-old using a self-report time diary for every 10 minutes to derive total sum of minutes per day per type of PA. The dataset categorised PA into six different activities (a) jogging, running, walking, hiking; (b) team ball games (e.g. football, hockey); (c) swimming and other water sports; (d) cycling; (e) individual ball games (e.g. tennis, badminton); (f) other exercise and sports, dancing, keeping fit, skiing, gymnastics). All types were included to represent PA in the model as previous studies suggest that complexity of PA is linked to self-regulation differently across childhood [2, 22]. Category (b) and (e) are considered open-skilled whilst the remaining categories are classified as closed skill sports [22, 57]. Type of PA at age 7 was not available due to self-report of this nature being too complex for this age group. PA was not available at age 11 as the cohort study at this age focussed on issues relevant to children entering adolescence such as risky behaviours.

**Academic achievement.** Previous studies suggest that self-regulation predicts academic achievement across several subject areas [15, 58, 59], as such subjects areas are combined to form one outcome variable in the hypothesised model. Multiple methods were used to measure achievement as one outcome variable at each time point. This included standardised achievement tests collected at 7-years-old and teacher reports at 11-years-old to overcome weakness of bias and scope of knowledge captured on standardised tests [60, 61].

At age 7, the British Ability Scales II word reading subtest was used to assess literacy and the Shortened version of the National Foundation for Education Research Standard Progress in Maths test for maths [62]. Both subtests are reliable and valid for use in England [63]. The word reading test involves naming words presented on a card. The maths test involves understanding shape, space and quantities and making simple calculations.

At age 11, core subjects (English language, mathematics, science, information and communication technology) and non-core subjects (music, art and physical education) were included in the analysis from teacher reports. We chose to include non-core subjects as it was expected that physical education might be correlated with the PA observed variable. Furthermore, research [64] suggest that characteristics of jobs to consider regarding the future labour market landscape include creativity. Teacher evaluated children using a five-point scale: "well below average", "below average", "average", "above average", "well above average". Academic achievement was not available at age 14 as the cohort study at this age focussed on issues relevant to young people's lives today including; mental health and wellbeing; occupational aspirations; language development.

**Socioeconomic status.** To incorporate risk factors in the model, we used the definition of SES encompassing several characteristics, including poverty, social status, family socioeconomic position, family atmosphere. Previous measures developed by Sammons et al. [65] were adapted to build an index of multiple disadvantages that are based on Indices of Cumulative

Risk. The data was collected through parental self-report questionnaire, assessed between 0–5 years of age. See Table 1.

**Control variables.** As additional measures, prior achievement and cognitive ability are included because they are commonly thought to predict outcomes [13, 71]. Prior achievement was assessed at the age of five using The Foundation Stage Profile, which was completed by each participant's teacher and is part of the statutory framework in England. This instrument covers six areas of learning, as a set of 13 assessment topics, each of which has nine points. The first three points describe a child who is still progressing towards the achievements described in the early learning goals. The next five points are drawn from the early learning goals them-selves. The final point in each scale is awarded when a child has exceeded expectations of the learning goal. Learning goals include, personal, social and emotional development; communi-cation, language and literacy; mathematical development; knowledge and understanding of the world; creative development; and physical development [72]. A range of academic adjust-ment measures, including creativity, have been included as part of the construct for prior aca-demic achievement. It has been suggested that creativity is a key factor for doing well in life [73] and benefits academic outcomes [74].

Cognitive ability was assessed at 5-years-old by evaluating verbal and non-verbal abilities [67]. The assessment was carried out by utilising the naming vocabulary subtest (verbal) and pattern construction with picture similarities subtests (non-verbal) from the British Ability Scales II [62]. The instrument assesses both cognitive ability and academic achievement. Both sub-tests can be interpreted individually, with the achievement sub-test not constructed to measure cognitive ability [62, 63] and are reliable and valid for use in United Kingdom England [63]. All scores were combined and then converted into t-scores [62]. The vocabulary subtest involves the child reading aloud a series of words presented on a card increasing with difficulty. The picture similarities test measures non-verbal reasoning and spatial visualisation and involve the ability to replicate a pattern presented on a card using flat foam squares or plastic cubes.

## Data processing and analyses

Data were analysed in R (v.3.3.2) using an additional package, lavaan [75] to run confirmatory factor analyses, measurement invariance, and the SEMs. A detailed summary of the data screening procedures and R scripts are available openly at https://osf.io/fyn5v/.

**Table 1. Summary of risk factors five-year olds or younger.**

| Characteristic | Demographic Data Collected |
|---|---|
| **Child** | First language spoken (not English) |
| | Pre-maturity or low birth weight (< 2.5kg or < 36 weeks) |
| **Economic** | Household income based on OECD scaling [66] |
| | Housing tenure (owned/mortgaged, privately renting, socially renting or others [67] |
| | Workless household |
| **Social** | Maternal age at birth (< 20 years of age) |
| | Single-parent household |
| | Maternal psychological distress [68, 69] |
| | Highest education status of parents (low: GCSE grades D–G or below; high: GCSE grades A*–C, A-Level, diploma and degree). |
| **Family** | Occupation status of both carers based on the Goldthorpe classification structure [70] |
| | Large family (> three siblings) |

Prior to testing the hypothesised model preliminary steps were taken, including standardising ability scores and confirmatory factor analyses to determine the factor structure of the measurement models. Multivariate outliers were detected using the Mahalanobis distance method ($\alpha > 0.05$) [76]. 280 outliers were excluded from the analyses. A comparison with and without outliers showed that model fit indices were quite similar, to avoid biases in the results only those excluding the outliers are reported here [77]. Multicollinearity was considered in this study and none was detected. Finally, longitudinal measurement invariance was measured across time for both items of self-regulation [15]. All models had an acceptable fit and met strict measurement invariance across time. See S4 Table.

All fit statistics were considered, including $\chi 2$, Comparative Fit Index (CFI), Tucker–Lewis Index (TLI), Root Mean Square Error of Approximation (RMSEA) Standardized Root Mean Square Residual indices (SRMR) were used, and a close fit view was considered to find an acceptable fit for each model tested by applying a two-index view [78]. A strict model, fixed factor loadings, thresholds and residuals, were used to address the research questions.

To evaluate each hypothesised model, the weighted least squares mean and variance-adjusted estimator was applied [79]. Negative test statistical results in the model indicate worse behavioural or emotional problems.

The inclusion criteria for the study includes children who wore a PA monitor at age 7 and surveyed in England (n = 4,043). Data based on the inclusion criteria at each time point are presented in the S1 Table. The dataset contained missing data due to natural attrition. The problem of missing data was resolved by implementing multiple imputations using Multivariate Imputation by Chained Equations as the model contained categorical data [80]. Robustness of the results was confirmed with five multiple imputed datasets [81].

## Results

### Characteristics of study sample

A majority of the participants where white (82%), 4.1% were Pakistani, 3.5% were of mixed race, 3.2% were Indian, 1.6% were Bangladeshi, 2% were Black African and 1.3% were Black Caribbean. The gender split was equal amongst boys and girls (50%). The sample under analyses comprised 14% of children whose first language spoken was not English, 10% who came from large families (> 3 children), 10% from workless households, 10% who were born premature, 10% who were part of a single parent household and 37% of mothers experiencing psychological distress. The average maternal age at birth was 29.84 years (SD = 5.47). Approximately 16% of main carers had no education status with the remainder holding GCSE grades A*–C (General Certificate of Secondary Education which is equivalent to a USA high school diploma), A-Level, diploma and degree. The housing tenure of families was owned or mortgaged (72.9%), privately renting (6.4%), socially renting (17.6%) or other (2.1%). The occupational status of families was managerial and professional (40.4%), intermediate occupation (14.8%), small employers (12.3%), supervisory and technical occupation (8.6%), or semi-routine and routine occupations (24%) from lowest to highest. The family income quantiles were 1st (14.4%), 2nd (16.6%), 3rd (20.8%), 4th (21%), and 5th (24.3%) from lowest to highest level.

### Descriptive statistics

Means, standard deviations, effect sizes and ranges for all variables are presented in the S1 Table. The reliability of emotional regulation (age 7: $\omega$ = .76; age 11: $\omega$ = .70; age 14: $\omega$ = .72) and behavioural regulation (age 7: $\omega$ = .75; age 11: $\omega$ = .65; age 14: $\omega$ = .73) were considered acceptable for all time points [82]. The data did not meet the assumptions of univariate and multivariate normality. The latter was not regarded as an issue because robust estimators and

multiple imputations [83] were used in this study to handle data that are not multivariate normal.

## Preliminary analyses

Confirmatory factor analyses was used to test the measurement model (see Table 2). Observable variables for emotional regulation from the CSBQ have been excluded as suggested by the initial round of confirmatory factor analyses. The fit indices indicate a good fit, with the exception of $\chi 2$. This test statistic becomes inflated when there are large samples [84] and was not considered when determining goodness of fit.

## Effects of physical activity on self-regulation

Structural equation modelling with latent variables were used to examine the effect PA had on emotional and behavioural regulation with and without controlling for SES. For the first research question, path estimates indicated a good fitting path model (see Fig 1 and Table 3) with PA positively and significantly predicting emotional regulation at ages 7 years ($\beta = .67$, $p < .001$), 11 years ($\beta = .68$, $p < .001$) and 14 years ($\beta = .31$, $p < .001$). However, the same was not true for behavioural regulation. Here, PA had a significant negative link to behavioural problems at 7 years ($\beta = -.54$, $p < .001$) and 11 years ($\beta = -.57$, $p < .001$), but was not statistically significant at 14 years ($\beta = -.04$, $p = .22$). When accounting for SES, the link between PA and both constructs of self-regulation at 7 years became negligible. However, path estimates revealed that after controlling for SES, PA significantly and negatively predicted emotional regulation at 11 years ($\beta = -.26$, $p < .001$) and 14 years ($\beta = -.30$, $p < .001$). On the other hand, PA had a negligible effect on behavioural regulation at 11 years ($\beta = -.02$, $p < .001$) and 14 years ($\beta = .00$, $p = .11$).

Specifically, environment of PA (indoor or outdoor) did not significantly predict emotional or behavioural regulation at age 7 years. Different types of categories of PA were factored into the model at 14 years (individual ball sports, swimming, running, team ball sports, other forms such as dancing). A combination of closed and open-skilled sports (swimming ($\beta = .48$, $p < .001$) and team ball games ($\beta = .27$, $p < .001$)) predicted both emotional and behavioural regulation, but the other types of PA were not statistically significant. See S1 Fig.

**Table 2. Confirmatory factor analysis measurement model fit indices.**

| Ages and Measures | $\chi^2$ (df) | RMSEA (90% CI) | CFI | TLI | SRMR |
|---|---|---|---|---|---|
| **0–5 years** | | | | | |
| Risk Factors | 258.14 (20) | .06 (.05 -.06) | .97 | .96 | .05 |
| **7 years** | | | | | |
| Emotional Regulation | 134.74 (5) | .08 (.07 -.09) | .98 | .96 | .05 |
| Behavioural Regulation | 20.25 (2) | .05 (.03 -.07) | .99 | .98 | .03 |
| Physical Activity | .00 (3) | .00 (.00 -.00) | 1.00 | 1.00 | 0 |
| Academic Achievement | 993.64 (20) | .11 (.10 -.12) | .97 | .96 | .08 |
| **11 years** | | | | | |
| Emotional Regulation | .00 (0) | .00 (.00 -.00) | 1.00 | 1.00 | 0 |
| Academic Achievement | 67.20 (14) | .03 (.02 -.04) | 1.00 | 1.00 | .03 |
| **14 years** | | | | | |
| Emotional Regulation | .00 (3) | .00 (.00 -.00) | 1.00 | 1.00 | 0 |
| Physical Activity | 110.42 (14) | .04 (.04 -.05) | .83 | .76 | .04 |

*Note.* All $\chi^2$ $p$ values were < .001.

RMSEA, root mean square error of approximation; CI, confidence interval; CFI, comparative fit index; TLI, tucker–lewis index; SRMR standardised root mean square residual.

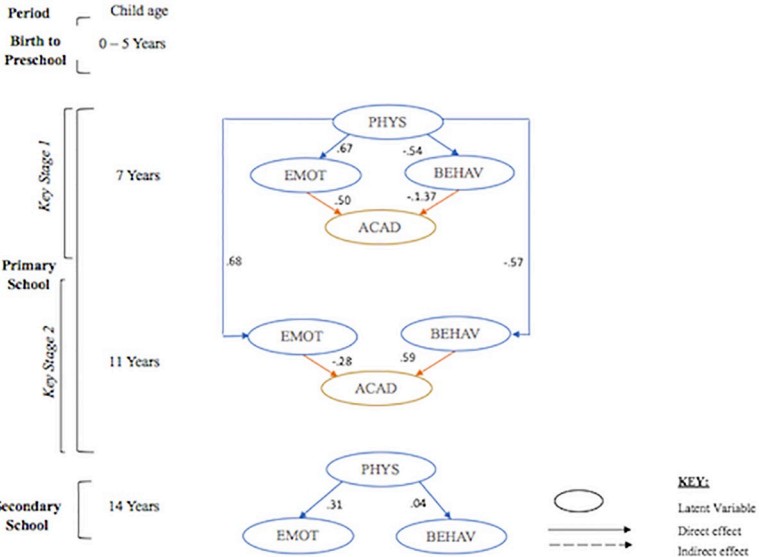

**Fig 1. Model without socioeconomic status.** PHYS, physical activity; EMOT, emotional regulation; BEHAV, behavioural regulation; ACAD, academic achievement.

### Effects of physical activity on self-regulation and academic achievement

The second purpose of this study was to identify the developmental effect of PA through the two aspects of self-regulation on academic achievement. Therefore, we predicted standardised tests and teacher evaluations of academic achievement in all subjects at age 7 and 11 using an index of PA at age 7 and emotional and behavioural regulation at age 7 and 11. The indirect effect of PA at age 7 on academic achievement was significantly positive through emotional

**Table 3. Predictions with and without socioeconomic status.**

|  | Without socioeconomic status | With socioeconomic status |
|---|---|---|
|  | *β* [95% CI] | *β* [95% CI] |
| **Effects on Self-Regulation** |  |  |
| Physical Activity (age 7) → Emotional Regulation (age 7) | .67 [.84, .96] | -.21 [-.35, -.24] |
| Physical Activity (age 7) → Behavioural Regulation (age 7) | -.54 [-.61, -.70] | -.04 [-.15, .04] |
| Physical Activity (age 7) → Emotional Regulation (age 11) | .68 [.88,.99] | -.26 [-.43, -.32] |
| Physical Activity (age 7) → Behavioural Regulation (age 11) | -.57 [-.73, -.64] | -.02 [-.12, .05] |
| Physical Activity (age 14) → Emotional Regulation (age 14) | .31 [.25, .39] | -.30 [-.38, -.28] |
| Physical Activity (age 14) → Behavioural Regulation (age 14) | .04 [-.02, .10] | .00 [.00, .00] |
| **Indirect Effects** |  |  |
| Physical Activity (age 7) → Emotional Regulation (age 7) → Academic achievement (age 7) | 1.77 [2.30, 2.63] | 2.61 [2.76, 3.23] |
| Physical Activity (age 7) → Behavioural Regulation (age 7) → Academic achievement (age 7) | -.10 [.58, .81] | -.99 [-.30, -.03] |
| Physical Activity (age 7) → Emotional Regulation (age 11) → Academic achievement (age 11) | .21 [.32, .57] | -.18 [-1.08, -.73] |
| Physical Activity (age 7) → Behavioural Regulation (age 11) → Academic achievement (age 11) | 1.10 [2.19, 2.42] | 1.55[3.96, 4.61] |

*Notes.* Strict measurement invariance was imposed. Model 1 without socioeconomic status ($\chi^2$ = 5,970.50, df = 358, $p$ < .001; CFI = .99, TLI = .99, RMSEA = .06, SRMR -.05), with socioeconomic status ($\chi^2$ = 7,445.41, df = 561, $p$ < .001; CFI = .99, TLI = .99, RMSEA = .06, SRMR = .05), Model 2 without socioeconomic status ($\chi^2$ = 2,676,68.27, df = 703, $p$ < .001; CFI = .96, TLI = .96, RMSEA = .21, SRMR = .16), with socioeconomic status ($\chi^2$ = 2,696,548.88, df = 1,035, $p$ < .001; CFI = .95, TLI = .95, RMSEA = .18, SRMR = .14).

CI, confidence interval; df, degrees of freedom; CFI, comparative fit index; TLI, tucker–lewis index RMSEA, root mean square error of approximation; SRMR standardised root mean square residual.

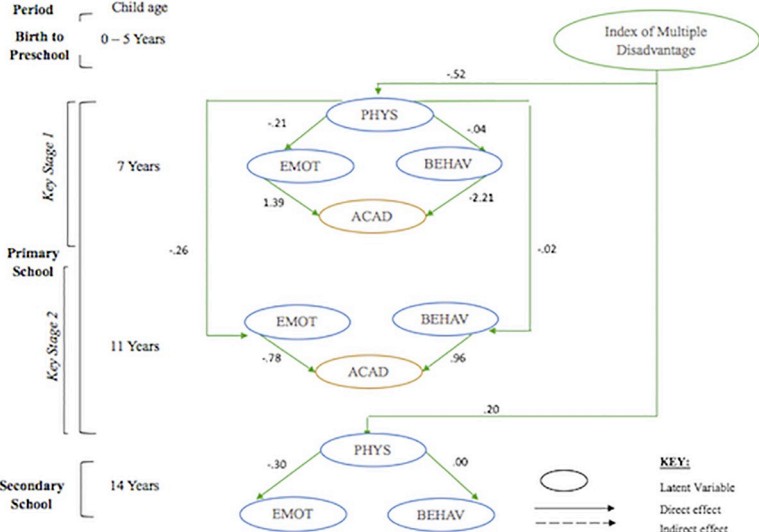

**Fig 2. Model with socioeconomic status.** PHYS, physical activity; EMOT, emotional regulation; BEHAV, behavioural regulation; ACAD, academic achievement.

regulation at 7 years (β = 1.77, p < .001) and behavioural regulation at 11 years (β = 1.10, p < .001), even after controlling for SES.

An examination of the correlation matrix (see S2 and S3 Tables) suggest these measures had a high correlation between behavioural regulation and PA at both time-points. Additionally, SES index and control variable cognitive ability and prior academic achievement at age 5 were also highly correlated. Fig 1 present the results of the structural model indicating the relationship between latent factors physical activity, emotional regulation and behavioural regulation in the short term at age 7 and 14, in the long term from age 7 to 11 and whether this association is linked to academic achievement. The model was also tested with SES factors which is shown in Fig 2.

## Discussion

To our knowledge, this is the first study to explore the both direct and indirect effects among PA, self-regulation and academic achievement longitudinally from childhood into early adolescence that factors in multiple indices of risk and an objective measure of PA. Our analyses, drawn from a large cohort sample has two main findings. First, there is a positive link between PA and emotional (not behavioural) regulation both concurrently and longitudinally across all three time points, 7-years-old, 11-years-old and 14-years-old. The relationship was negative for emotional regulation and negligible for behavioural regulation when controlling for SES. Second, across two time points (due to data availability), PA positively predicted academic achievement through emotional self-regulation for 7-year-olds and behavioural regulation in 11-year-olds when controlling for prior achievement and cognitive ability at age five. The impact of this relationship was more pronounced when controlling for SES.

### Physical activity and self-regulation

This study offers further insight into the specific aspects of self-regulation that are linked to PA during childhood and adolescence. We found that PA across childhood and adolescence positively predicted emotional regulation skills, with the opposite being true for behavioural regulation. Behavioural regulation had a negligible positive relationship at 14-years-old. Our

findings complement prior studies identifying emotional regulation as playing a larger role than behavioural regulation in early and middle childhood [15] and that PA does predict an aspect of self-regulation (e.g. [24]). This could be because children that can manage their emotions are better placed to regulate their behaviour (Blair, 2002; Ng et al., 2015). Factors of self-regulation are open to environmental influences because of the timing of brain plasticity. The brain areas associated with emotional and behavioural regulation (the amygdala and prefrontal cortex) develop throughout childhood and adolescence.

Alternatively, it is possible that the relationship between PA and behavioural regulation is not as important in early and middle childhood because the adults in children's lives are more forgiving of behavioural misunderstandings and help them manage their behaviour. One study investigated the differences in learning environments between primary and secondary schools identified a deterioration in the quality of teacher-student interaction on factors such as being helpful, friendly and understanding of freedom behaviours [85]. Furthermore, when entering adolescence, children become increasingly independent and rely more on peers and less on adults and this might show in their behaviour towards adults [86].

Evidence has shown that self-regulation is greater as risk lowers and that children farthest behind will experience greater benefits from activities that affect self-regulation [25]. One study examining self-regulation for children who experienced risk and PA in early childhood had mixed findings [16]. Our findings showed that PA was not related to either factors of self-regulation after accounting for SES factors in early childhood. This could be due to poorer self-regulators being less likely to take part in PA supporting the bi-directional association of PA and self-regulation [24]. Given the negligible or negative effect PA had on self-regulation longitudinally when controlling for SES, teachers should take a holistic approach when considering social disadvantage.

The observation that certain types of PA have more of an effect on self-regulation is consistent with empirical studies [25] and childhood studies [22, 24]. Specifically, individual sports have been shown to effect self-regulation in early childhood, whereas team sports have had no effect [24]. In adolescence, only one type of closed and open-skilled PA (swimming and team ball sports respectively) was associated to self-regulation. Team sports involve self-regulation challenge to improve them (sustained attention, switching roles, overriding maladaptive impulses, etc.) and it is also possible that swimming might also require careful sustained attention and control to overcome lapses in form. Diamond and Ling [25] suggested that it is not PA alone that affects mental processes in children [87] but a combination of PA and character development that enhances self-regulation, for example, martial arts [88].

## Physical activity, self-regulation and academic achievement

Children that struggle to self-regulate their behaviour could be more impulsive and inattentive and may have more difficulties with learning, potentially affecting their academic achievement [89]. Prior studies indicate that a student's academic achievement is associated to self-regulation skills [15, 90–94]. Although most of these studies used parental assessments of self-regulation, recent meta-analytic research confirmed the relationship even when teacher assessments and task based assessments of self-regulation in relation to academic achievement produce comparable results, with both having larger effect sizes than parental assessments [95].

Others have also shown that it isn't only self-regulation in childhood that predicts academic achievement, but that real-world outcomes, such as well-being, education and labour market outcomes, improve when self-regulation improves [24, 58, 60, 71, 96, 97].

Past literature also indicates a direct association between PA and academic achievement (e.g., [35]) and an indirect association through self-regulation (e.g. [16]).

The current study has expanded on previous findings by identifying elements of self-regulation, which are important to academic outcomes at different time-points that are linked to PA [16]. Findings indicate that PA positively predicts emotional regulation skills resulting in higher achievement throughout early primary school with behavioural regulation resulting in higher achievement in middle childhood. The impact was more pronounced when controlling for SES. In addition, children that are farthest behind could experience greater benefits from activities that are linked to self-regulation [25, 98]. Child poverty has been shown to be associated with poorer self-regulation skills and academic attainment in childhood. In 2016, one child in seven in OECD countries were living in poverty [99]. In our own country, the United Kingdom, The Institute of Fiscal Studies forecast that child poverty would increase by 3% by the end of the present parliament [100]. Children from low socioeconomic status (SES) families risk starting school with fewer skills or assets, occurring as early as kindergarten and continuing throughout school [101]. Parents with low SES may not have the time or money [102] to provide a suitable environment for allowing their children to develop skills such as self-regulation [103]. Previous studies have suggested that the attainment gap that exists for disadvantaged children is the result of poor self-regulation skills in early childhood [13].

Another reason could be the contextual differences in PA participation. Aggio et al. [41] investigated the link between PA at 7-years-old and cognition at 11-years-old and discovered that frequency of sports club participation led to higher cognitive scores. There is a clear distinction between organised PA and PA during leisure time as the former tends to be goal-orientated and strategic [104]. This contextual difference might be true for academic achievement. The present study only took different types of PA into consideration in adolescence where academic outcomes were not available.

Alternatively, the results can be attributed to academic achievement being measured by teacher judgements at 11-years-old and not a combination of standardised tests and teacher judgements as was done at 7-years-old. Behaviour is directly observable by teachers who may become biased when evaluating children's performance. Students are judged more positively by their teachers when their personalities are similar [105]. There are also longitudinal effects of teacher judgement and treatment of students on academic outcomes. Teacher judgements directly related to students' future achievement and supported the self-fulfilling prophecy model of Brophy and Good [106].

The role that PA plays on educational outcomes found in the present study enhances the current literature by shedding light on aspects of self-regulation that are associated with academic achievement. Emotional regulation encompasses aspects of being on task and maintaining attention, which may be the most problematic in the attainment gap. The findings lead to some practical guidance: an activity that influences academic achievement can provide a partial bridge to the attainment gap early or helps children catch up later [25].

## Strengths and limitations

This study suggests a link between PA, multi-dimensional view of self-regulation and academic achievement. This study has several strengths. First, its use of multiple time points to identify developmental patterns. Second, the exploration of directional processes using SEM as the set of latent variables represented hypothetical constructs using different instruments. Third, consistent with risk and resilience research, this paper adopted an epidemiological approach applying multiple disadvantage factors in relation to cumulative risk. Finally, the combination of constructs in the child's environment and individual differences, particularly physical activity and self-regulation, allows us to have a deeper understanding of how a combination of factors contribute to developmental trajectories and how this can translate to interventions and improvement in the teaching environment.

However, there are several limitations. First, this study was longitudinal with multiple measures and informants with a correlational design, which has the shortcomings of methodological design, mainly lacking causality. Second, the data on the frequency of PA was not available. The combination of intensity, frequency and duration better represents the construct PA as whole and is likely to predict self-regulation and academic achievement for children who are disadvantaged as they have lower participation rates. Third, by addressing omitted variable bias, this study would have benefited from including variables that were not available in the data set, such as unbiased assessment of self-regulation using children's self-reports from an early age, factoring in the content of the PA such as dancing, team sports and martial arts in early childhood [107] and teacher assessment of self-regulation instead of parental assessment at age 14. Teacher assessment of self-regulation and task based assessments in relation to academic achievement produce comparable results but not parent assessment of self-regulation [95]. Lastly, understanding labour market success by including this data time-point would contribute to understanding how the aspects of the school environment and the individual translate into adulthood. Although there is evidence that self-regulation is crucial for success in the labour market [108], because of the geopolitical landscape and speed of technological progression, the labour market has already changed and may change further. To date, this cohort study has not reached this data time-point.

## Implications for policy and practice in education

Psychologists and educators should be sensitive to early risk factors and the effect of different experiences on outcomes. Although the findings did not support previous literature in relation to the link between PA and self-regulation in its entirety, by considering SES and individual experiences throughout children's school life, the results highlighted the complex nature and the difficulty of disentangling elements related to early risk.

The findings suggest that the focus on PA that policymakers are adopting is the right one. However, in recent years the education system in the UK has prioritised literacy and numeracy results as the silver bullet to employability and developing the 'knowledge economy', side-lining PA. Educators understanding that PA is not for only for the body but for the mind as well, dissolving the Cartesian divide between the two. To help, the PA guidelines should also outline the process and outcome measures so that schools can easily implement them.

At a local level, councils can create safe, family-friendly indoor and outdoor environments to ensure that PA takes place [109]. At the school level, educators could engage with sports clubs outside school, particularly swimming and ball sports, to create targeted sports programmes taking place in any environment. Another option for educators is to adopt a holistic approach [110] by adapting the school environment and the curriculum and by including the family where possible for school-based interventions, particularly for those experiencing early risk [111]. Taking a holistic approach requires initiative from the school leadership team and appropriate training for all staff [112]. To realise an effect, the amount and intensity of PA undertaken, interventions should run for an entire school year [113]. With any of the actions outlined, educators should ensure that the targeted behaviours are clear before starting. That is, an emotional component in early to middle childhood and behavioural regulation in the latter years.

## Future directions and conclusion

Identifying pathways in early childhood that have an impact on academic, emotional and social outcomes are critical for later development and becoming a well-adjusted adult. The present study examined the relationship amongst PA, self-regulation in a multi-dimensional

approach and educational outcomes when controlling for background factors across the children's time at school. Consistent with risk and resilience research, this study adopted an epidemiological approach applying multiple disadvantage factors in relation to cumulative risk. The combination of constructs in the child's environment and individual differences allowed a deeper understanding of how a combination of factors contributes to developmental trajectories and how it can result into interventions and improvements in the teaching environment. This study showed that emotional regulation is linked to PA in early childhood to subsequently affect academic achievement, suggesting that the focus should be on attention and not behaviour. For disadvantaged children, educators should promote government policy on PA to indirectly influence the attainment gap, however other interventions are needed to directly influence self-regulation.

From a methodological perspective, the SEM approach that was used has been proven useful in testing hypotheses about potential relationships linking early risk, PA, self-regulation and outcomes while controlling for pre-academic achievement and cognitive ability. However, application issues such as multicollinearity need to be considered and acknowledged. Furthermore, as the present study used secondary data, limitations in relation to the availability of variables and access to a second data set to comply with SEM best practices were also problematic. Although omitted variables can be a general limitation of secondary data analysis, the results may open the window to inform the design of future research. Future research, for example, could include a range of other academic measures which are going to be important for the future labour market such as creativity and critical thinking to help us understand how these factors contribute to labour market success [64]. Other opportunities for future research include employing a cross-cultural approach applying measurement invariance to understand the same constructs in a different environment. Cultural nuances may also shape the educational environment and consequently, children embedded in another education system may have different developmental trajectories if those factors had not been there. Exploring the proposed model with another sample may mitigate weaknesses in the analysis and address SEM best practice.

## Supporting information

**S1 Table. Descriptive statistics.**
(DOCX)

**S2 Table. Correlation without academic achievement.**
(DOCX)

**S3 Table. Correlation with academic achievement.**
(DOCX)

**S4 Table. Fit statistics for longitudinal measurement invariance.**
(DOCX)

**S1 Fig. Complete model with physical activity, emotional regulation, behavioural regulation and risk.** PA, physical activity; ER, emotional regulation; BR, behavioural regulation; PHYS, physical activity; EMOT, emotional regulation; BEHAV, behavioural regulation. The parameters on the left side of the slash are without controlling for SES, on the right controlled for SES.
(TIF)

**S2 Fig. Sample selection criteria.** Inclusion criteria for sub-sample from the Millennium Cohort Study London: Centre for Longitudinal Studies, Institute of Education, University of

London. Maximum number of participants in sample.
(TIF)

## Acknowledgments

The authors would like to thank the UK data service, The Faculty of Education, Cambridge University and Ulrich Schroeders of the Universität Kassel, Department of Psychology for supporting open science and sharing the syntax to test for configural, strong, and strict longitudinal MI with categorical data as part of his co-authorship of his paper on the development of emotional and behavioural self-regulation and their effects on academic achievement in childhood [15].

## Author Contributions

**Conceptualization:** Fotini Vasilopoulos.

**Formal analysis:** Fotini Vasilopoulos.

**Investigation:** Fotini Vasilopoulos.

**Methodology:** Fotini Vasilopoulos.

**Software:** Fotini Vasilopoulos.

**Supervision:** Michelle R. Ellefson.

**Validation:** Fotini Vasilopoulos.

**Writing – original draft:** Fotini Vasilopoulos.

**Writing – review & editing:** Michelle R. Ellefson.

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
