## [Decision Letter · Decision Letter 0]

22 Dec 2020

PONE-D-20-31387

Investigation of the associations between physical activity, self-regulation and educational outcomes in childhood

PLOS ONE

Dear Dr. Vasilopoulos,

Thank you for submitting your manuscript to PLOS ONE. After careful consideration, we feel that it has merit but does not fully meet PLOS ONE’s publication criteria as it currently stands. Therefore, we invite you to submit a revised version of the manuscript that addresses the points raised during the review process.

I agree with the reviewers that the manuscript, mainly the Introduction, is too long. Therefore, I recommend summarizing it. Authors might consider avoiding repetitions and restructuring the manuscript by moving some paragraphs from the Introduction to the Discussion. Another important recommendation is to increase fluency through the manuscript.

It is necessary to follow these two rules: 

1) "Data analysis" belongs to the Methods section. 

2) Figures and tables should be self-explanatory. Therefore, please provide the full phrase for abbreviations in the figure's legend or the table's footnote. 

We look forward to receiving your revised manuscript.

Kind regards,

Thalia Fernandez, Ph.D.

Academic Editor

PLOS ONE

Journal Requirements:

2.We note that you have indicated that data from this study are available upon request. PLOS only allows data to be available upon request if there are legal or ethical restrictions on sharing data publicly. For more information on unacceptable data access restrictions, please see http://journals.plos.org/plosone/s/data-availability#loc-unacceptable-data-access-restrictions.

3. We note you have included a table to which you do not refer in the text of your manuscript. Please ensure that you refer to Table 1 in your text; if accepted, production will need this reference to link the reader to the Table.

Reviewers' comments:

Reviewer's Responses to Questions

**Comments to the Author**

1. Is the manuscript technically sound, and do the data support the conclusions?

Reviewer #1: Yes

Reviewer #2: Yes

Reviewer #3: Yes

2. Has the statistical analysis been performed appropriately and rigorously? 

Reviewer #1: I Don't Know

Reviewer #2: Yes

Reviewer #3: I Don't Know

3. Have the authors made all data underlying the findings in their manuscript fully available?

Reviewer #1: Yes

Reviewer #2: Yes

Reviewer #3: No

4. Is the manuscript presented in an intelligible fashion and written in standard English?

Reviewer #1: Yes

Reviewer #2: Yes

Reviewer #3: Yes

5. Review Comments to the Author

Reviewer #1: This cohort study assesses the relationship between physical activity, self-regulation, and educational outcomes in students 7 to 14 years old. The authors have put in the great effort; however, it needs a major revision before acceptance.

Abstract:

Please provide years of study.

Introduction:

Line 38: Please provide the full name before the abbreviation “high-SES.”

The introduction is lengthy; please summarize it to what we know, what we don’t know, and the study aim. Please move the “Relations between Self-Regulation and Academic Achievement” and “Physical Activity and Childhood Outcomes” parts of the introduction to the discussion section.

Method:

Line 240: Please provide the full name before the abbreviation “MCS.”

Please move this part to the results section: “The sample under analyses comprised 14% of children whose first language spoken was not English, 10% who came from large families (> 3 children), 10% from workless households, 10% who were born pre-mature, 10% who were part of a single parent household and 37% of mothers experiencing psychological distress. The average maternal age at birth was 29.84 years (SD = 5.47). Approximately 16% of main carers had no education status with the remainder holding GCSE grades A*–C, A-Level, diploma and degree. The housing tenure of families was owned or mortgaged (72.9%), privately renting (6.4%), socially renting (17.6%) or other (2.1%). The occupational status of families was managerial and professional (40.4%), intermediate occupation (14.8%), small employers (12.3%), supervisory and technical occupation (8.6%), or semi-routine and routine occupations (24%) from lowest to highest. The family income quantiles were 1st (14.4%), 2nd (16.6%), 3rd (20.8%), 4th (21%), and 5th (24.3%) from lowest to highest level.”

Did the authors measure the amount of physical activity during transportation for the subjective measurement section? Or only accessed the activity during sports participation?

Line 368: Please refer to table one in the text before its presentation.

Results:

Lines 401 to 434: Please move “Data Processing and Analyses” to the method section.

Table 2: Please provide the full phrase for abbreviations used in the table (RMSEA, CFI, TLI, and SRMR) in the table's footnote.

Discussion:

The mentioned parts of the introduction can be presented in the discussion section.

Reviewer #2: Investigation of the associations between physical activity, self-regulation and

educational outcomes in childhood - The manuscript is well written. The study has been designed with scientific rigor from its inception stage to final analysis. The conclusions drawn has answered the Research Question.

Reviewer #3: Review

Manuscript Number: PONE-D-19-09923

Article Type: Research Article

Full Title: Investigation of the associations between physical activity, self-regulation andeducational outcomes in childhood

Short Title: Physical activity, self-regulation and educational outcome

Corresponding Author: Fotini Vasilopoulos, Med

Birkbeck University of London, UNITED KINGDOM

General notes

Thank you for this good manuscript. The topic of this paper is important and public health relevance is high. I appreciate the amount of work in the manuscript. While this is an interesting article, I have some comments.

Two Universities were given (University of London & University of Cambridge. Which one is right?

The length of the manuscript is 35 pages, this is too long. Although manuscripts can be any length at PLOS ONE, I recommend shortening the manuscript. There are numerous repetitions in various places in the manuscript. So there is a lot of potential for cuts. Please see line 359/360 (example): This information has already been given and it is therefore not necessary to repeat it.

Introduction

The introduction length is 10 pages. There are numerous repetitions in the introduction section. Please try to shorten this section.

In general, the manuscript (In particular the introduction and the discussion section) would benefit from restructuring. Therefore, I recommend the following structure:

- Status quo in the literature & study situation

- Definitions and characteristics (self-regulation, PA)

- Then the 1) connection between PA and self-regulation, 2) PA and long-term outcomes and then 3) effects of PA on academic achievement

- Present your own study at the end of the introduction and avoid jumps in the introduction

I recommend using PA as a shortcut of physical activity in the whole manuscript.

How was self-regulation measured in the mentioned studies? Are the results comparable? Please complete in the manuscript.

Line 90: Please introduce the shortcut SES fort he first time

Line 39: Please complete the recommendations for PA for this age group.

Line 72: The bracket is missing

Line 74: Please give information why cognitive regulation is missing?

Line 83: Please complete which outcomes

Line 96: Please give some examples

Line 97: One word is missing here.

Line 228: Please add some details about the accelerometry (e.g. model name, number of axes, wearing time, wearing criteria, wrist or hip?

Line 223: Please add information about “MCS” in brackets

Line 247: or � of

Lines 248 – 260: I do recommend to represent the information in a table

Line 253: Please add information about „GCSE“

Line 266: The bracket is missing

Methods:

Why you only use 2 items of the SDQ scale? The SDQ consist of several scales, each of which forms a score. Why you didn’t use the score (5 items)?

The measurement of the self-regulation is carried out by teachers & parents (7 yr) or only parents (11, 14 yr). Why wasn't that done consistently? Are the results comparable? I recommend to add this part in the limitation and discussion section.

Line 291: Please add number of supplement

Supplement 1: Please adapt the items in order to the manuscript (first: emotional regulation, second: beh. regulation)

Lines 292/293: Please delete the blank lines

Lines 290/291: Please add number of supplement

Line 307: Please give more details about the accelerometer measurement (see my comment above)

Line 308: Please add more information about “Euclidean Norm Minus One” and please explain what does mg stand for

Line 311: Are the raw accelerometer data comparable with the Cut-Offs (7 & 11 yr)?

Line 313: The items you describe do not measure frequency of PA. Instead the described items measure enjoyment.

In the age of 14 years the academic achievement was not measured. Please add this fact to the discussion section.

Line 357: Please add information about the assessment of the SES. When and how was it assessed?

Table 1: Please add a reference in the manuscript (see table 1).

Results

In my opinion, the results section starts on line 435. The previous information relates to the methods (statistics).

Line 430: Please include the number of the addendum.

Line 437: Please include the number of the addendum.

Figure 1: Please give more information about figure 1. Explain the figure in more detail in the manuscript.

Line 529: Please try to specify the possible grounds.

I recommend to add strenghs of the study to the limitations part

I recommend to structure a short part “future directions and conclusion” as a separate part

There is also theoretical information in the results section. I recommend to limit yourself to the results. In the discussion, the results of your own study can be linked to the introduction.

Try to shorten and to structure the discussion section in the same order like the introduction section.

There is no sample description: I recommend to add a table which describes the sample (age, sex, ….)

There is no consort chart: I recommend to add a consort for transparency of the sample composition

Discussion

Discussion section should include information that cognitive ability only assessed at 5 year olds.

- PA was not assessed at 11 year olds.

6. PLOS authors have the option to publish the peer review history of their article (what does this mean?). If published, this will include your full peer review and any attached files.

Reviewer #1: No

Reviewer #2: No

Reviewer #3: No

---

## [Author Response · Author response to Decision Letter 0]

28 Jan 2021

EDITOR COMMENT: Thank you for submitting your manuscript to PLOS ONE. After careful consideration, we feel that it has merit but does not fully meet PLOS ONE’s publication criteria as it currently stands. Therefore, we invite you to submit a revised version of the manuscript that addresses the points raised during the review process.

I agree with the reviewers that the manuscript, mainly the Introduction, is too long. Therefore, I recommend summarizing it. Authors might consider avoiding repetitions and restructuring the manuscript by moving some paragraphs from the Introduction to the Discussion. Another important recommendation is to increase fluency through the manuscript.

RESPONSE: Thank you for coordinating the review process. We have revised the

manuscript and addressed Reviewer’s comments, including summarising and restructuring the introduction. In response to your concerns about the length and amount of information in the introduction, we have substantially moved some of the paragraphs to the Discussion based on Reviewer 1 recommendations (“Relations between Self-Regulation and Academic Achievement” 23 lines) and removed other lines (28 lines). Overall, we have shortened the introduction from 10 pages to 7 pages. Specifically, we now provide a more streamlined introduction section that is consistent with Reviewer 1 and 3 recommendations (see below for details). 

We note that Reviewer 3 has requested an overall reduction in the length of the manuscript (35 pages). We have reduced the length by 3 pages by reducing and streamlining the Introduction and making edits throughout the manuscript. Reviewer 3 requested additions to the paper which could reduce the length of the manuscript further. These include a new sub-heading in the Introduction (“Physical activity and long-term outcomes 9 lines), additional physical activity information in the “Measures” section (16 lines) and information about the “Strengths” of the study in the limitations section (10 lines).

General Comments:

It is necessary to follow these two rules: 

1) "Data analysis" belongs to the Methods section. 

2) Figures and tables should be self-explanatory. Therefore, please provide the full phrase for abbreviations in the figure's legend or the table's footnote. 

RESPONSE: We have moved the “Data analysis” section to the Methods section. We have updated the figure’s legend and table footnotes for full phrase abbreviations.

COMMENT: Please ensure that your manuscript meets PLOS ONE's style requirements, including those for file naming. 

RESPONSE: We have checked and attest that all formatting and style requirements have been met. 

COMMENT: We note that you have indicated that data from this study are available upon request. PLOS only allows data to be available upon request if there are legal or ethical restrictions on sharing data publicly. For more information on unacceptable data access restrictions, please see http://journals.plos.org/plosone/s/data-availability#loc-unacceptable-data-access-restrictions.

RESPONSE: The data that support the findings of this study are available from the UK Data Service repository. Restrictions apply to the availability of these data, which were used under licence for this study. The authors of this study are not the owners of this data and had to make a specific request to access it for the analyses reported here. Data are available https://beta.ukdataservice.ac.uk/datacatalogue/series/series?id=200003 with the permission of UK Data service repository. Detailed information about the processing of the data used in this manuscript can be found at https://osf.io/fyn5v/.

COMMENT: We note you have included a table to which you do not refer in the text of your manuscript. Please ensure that you refer to Table 1 in your text; if accepted, production will need this reference to link the reader to the Table.

RESPONSE: We added a sentence in “Measures” section subsection “Socioeconomic status” to refer to table 1. See line 304.

COMMENT: Please include captions for your Supporting Information files at the end of your manuscript, and update any in-text citations to match accordingly. Please see our Supporting Information guidelines for more information: http://journals.plos.org/plosone/s/supporting-information.

RESPONSE: We have included captions at the end of the manuscript for all supporting information files and have updated in-text citations. 

 

RESPONSE TO REVIEWER 1:

COMMENT: This cohort study assesses the relationship between physical activity, self-regulation, and educational outcomes in students 7 to 14 years old. The authors have put in the great effort; however, it needs a major revision before acceptance.

RESPONSE: We have revised our manuscript based on the comments as described below. 

COMMENT: Abstract: Please provide years of study.

RESPONSE: We have added the wording “a cohort of infants born in 2000–2001 in the United Kingdom” to the abstract. See line 14.

COMMENT: Introduction: Line 38: Please provide the full name before the abbreviation “high-SES.”

The introduction is lengthy; please summarize it to what we know, what we don’t know, and the study aim. Please move the “Relations between Self-Regulation and Academic Achievement” and “Physical Activity and Childhood Outcomes” parts of the introduction to the discussion section.

RESPONSE: We have added the full name socioeconomic status before “high-SES” at line 38. 

The editor and reviewer 3 have raised the same point with a different recommendation of structure. In response to your concerns about the length and amount of information in the introduction, we have substantially moved some of the paragraphs to the Discussion based on your recommendations (“Relations between Self-Regulation and Academic Achievement” 23 lines) and removed other lines (28 lines). Overall, we have shortened the introduction from 10 pages to 7 pages. Specifically, we now provide a more streamlined introduction section that is consistent with yours and Reviewer 3 recommendations. We have adopted the recommended sub-headings by Reviewer 3 and therefore kept one of the two sub-sections that you suggest to move to the discussion section. We have also followed your recommendations of “what we know, what we don’t know” throughout each sub-heading of the introduction. Based on Reviewer 3 comments we have included the sub-heading “PA and long-term outcomes” in the introduction and although it is important, this does not necessarily form part of the model or study so they are lines which could be excluded altogether. Please confirm whether this should be included or not.

COMMENT: Method: Line 240: Please provide the full name before the abbreviation “MCS.”

Please move this part to the results section: “The sample under analyses comprised 14% of children whose first language spoken was not English, 10% who came from large families (> 3 children), 10% from workless households, 10% who were born pre-mature, 10% who were part of a single parent household and 37% of mothers experiencing psychological distress. The average maternal age at birth was 29.84 years (SD = 5.47). Approximately 16% of main carers had no education status with the remainder holding GCSE grades A*–C, A-Level, diploma and degree. The housing tenure of families was owned or mortgaged (72.9%), privately renting (6.4%), socially renting (17.6%) or other (2.1%). The occupational status of families was managerial and professional (40.4%), intermediate occupation (14.8%), small employers (12.3%), supervisory and technical occupation (8.6%), or semi-routine and routine occupations (24%) from lowest to highest. The family income quantiles were 1st (14.4%), 2nd (16.6%), 3rd (20.8%), 4th (21%), and 5th (24.3%) from lowest to highest level.”

RESPONSE: We have removed the abbreviation and provided the full name Millennium Cohort Study. See line 180. We have moved the paragraph referred to above to the results section under the sub-title “Characteristics of Study Sample”. See lines 367-384.

COMMENT: Method: Did the authors measure the amount of physical activity during transportation for the subjective measurement section? Or only accessed the activity during sports participation?

RESPONSE: The Millennium Cohort Study accessed activity during sports participation only.

COMMENT: Method: Line 368: Please refer to table one in the text before its presentation.

RESPONSE: We added a sentence in “Measures” section subsection “Socioeconomic status” to refer to table 1 before its presentation. See line 304.

COMMENT: Results: Lines 401 to 434: Please move “Data Processing and Analyses” to the method section. Table 2: Please provide the full phrase for abbreviations used in the table (RMSEA, CFI, TLI, and SRMR) in the table's footnote.

RESPONSE: We have moved the “Data analysis” section to the Methods section. See lines 335 - 365. We have updated the figure’s legend and table footnotes for full phrase abbreviations.

COMMENT: Discussion: The mentioned parts of the introduction can be presented in the discussion section.

RESPONSE: We have moved and streamlined “Relations between Self-Regulation and Academic Achievement” parts of the introduction to the Discussion section under the heading “Physical Activity, Self-Regulation and Academic Achievement. See lines 520-533. We have kept with Reviewer 3 recommendations and kept “Physical Activity and Childhood Outcomes” in the Introduction.

RESPONSE TO REVIEWER 2:

COMMENT: Investigation of the associations between physical activity, self-regulation and educational outcomes in childhood - The manuscript is well written. The study has been designed with scientific rigor from its inception stage to final analysis. The conclusions drawn has answered the Research Question.

RESPONSE: We thank the reviewer for their kind words.

 

RESPONSE TO REVIEWER 3:

General notes

COMMENT: Thank you for this good manuscript. The topic of this paper is important and public health relevance is high. I appreciate the amount of work in the manuscript. While this is an interesting article, I have some comments.

RESPONSE: We thank the reviewer for their kind words. All comments are addressed below.

COMMENT: Two Universities were given (University of London & University of Cambridge. Which one is right?

RESPONSE: We have contacted the help centre and they have confirmed that

“If an author has multiple affiliations, enter all affiliations on the title page only. In the submission system, enter only the preferred or primary affiliation. Author affiliations will be listed in the typeset PDF article in the same order that authors are listed in the submission.”

 Both are correct: one is my current affiliation and one is where the work took place. We have amended the title page to reflect the guidance given from the help centre.

COMMENT: The length of the manuscript is 35 pages, this is too long. Although manuscripts can be any length at PLOS ONE, I recommend shortening the manuscript. There are numerous repetitions in various places in the manuscript. So there is a lot of potential for cuts. Please see line 359/360 (example): This information has already been given and it is therefore not necessary to repeat it.

COMMENT: Introduction: The introduction length is 10 pages. There are numerous repetitions in the introduction section. Please try to shorten this section.

In general, the manuscript (In particular the introduction and the discussion section) would benefit from restructuring. Therefore, I recommend the following structure:

- Status quo in the literature & study situation

- Definitions and characteristics (self-regulation, PA)

- Then the 1) connection between PA and self-regulation, 2) PA and long-term outcomes and then 3) effects of PA on academic achievement

- Present your own study at the end of the introduction and avoid jumps in the introduction

RESPONSE: The editor and reviewer 1 have raised the same point with a different recommendation of structure. In response to your concerns about the length and amount of information in the introduction, we have substantially moved some of the paragraphs to the Discussion based on Reviewer 1 recommendations (“Relations between Self-Regulation and Academic Achievement” 23 lines) and removed other lines (28 lines). Overall, we have shortened the introduction from 10 pages to 7 pages. Specifically, we now provide a more streamlined introduction section that is consistent with Reviewer 1 and your recommendations. We have adopted the recommended sub-headings and therefore kept some of the two sub-sections that reviewer 1 suggests to move to the discussion section.

Based on your comments we have included the sub-heading “PA and long-term outcomes” in the introduction and although it is important, this does not necessarily form part of the model or study so they are lines which could be excluded altogether. Please confirm whether this should be included or not. We have reduced the overall length of the manuscript (31 pages from 35 pages).

COMMENT: I recommend using PA as a shortcut of physical activity in the whole manuscript.

RESPONSE: We have updated the manuscript for this abbreviation.

COMMENT: How was self-regulation measured in the mentioned studies? Are the results comparable? Please complete in the manuscript.

RESPONSE: We have moved this section to the Discussion section based on Reviewer 1 comments under the sub-heading “Physical Activity, Self-Regulation and Academic Achievement” and the wording has significantly changed. See lines 520-533. This Discussion section now includes the following wording to address your point about comparability. “Although most of these studies used parental assessments of self-regulation, recent meta-analytic research confirmed the relationship even when teacher assessments and task based assessments of self-regulation in relation to academic achievement produce comparable results, with both having larger effect sizes than parental assessments” see lines 523-527.

Line 90: Please introduce the shortcut SES for the first time

RESPONSE: We have added the wording “socioeconomic status” at the relevant line 38.

Line 39: Please complete the recommendations for PA for this age group.

RESPONSE: The following sentence has been added “The United Kingdom government PA guidelines recommend that children ages 5 – 18 years of age should aim for an average of 60 minute of moderate intensity PA per day”. See lines 39-41.

Line 72: The bracket is missing

RESPONSE: We have added the bracket at the relevant line.

Line 74: Please give information why cognitive regulation is missing?

RESPONSE: We did not include cognitive regulation as part of this study because the relationship between physical activity and cognitive aspects of regulation (i.e. executive functions) has a) been studied extensively and b) development of social and emotional skills help with the transition into adulthood. Understanding the social and emotional aspects of regulation will allow future research to integrate the social and emotional findings with cognitive findings to create a developmental picture in a “natural setting’. We have added the sentence “We have chosen to focus on emotional and behavioural regulation skills as they help with the transition to adulthood so that findings can be integrated with cognitive research to create a developmental picture in a natural context.” to clarify this point in the paper at lines 153-156.

Line 83: Please complete which outcomes

RESPONSE: This section has been moved on recommendation of Reviewer 1.

We have included the following wording “real-world outcomes, such as adult health and wealth” at the relevant line. See lines 529-530. 

Line 96: Please give some examples

RESPONSE: This section has been moved on recommendation of Reviewer 1. We have included the following wording “well-being and labour market” at line 529-530.

Line 97: One word is missing here.

RESPONSE: This section has been moved on recommendation of Reviewer 1. It has been streamlined and the sentence has been removed.

Line 228: Please add some details about the accelerometry (e.g. model name, number of axes, wearing time, wearing criteria, wrist or hip?

RESPONSE: We have included this detail in the sub-section “physical activity” under the “Measures” section. The following sentences have been added “At age 7 the Actigraph GT1M uni-axial accelerometer was worn on the right hip for seven consecutive days during waking hours; and at age 14 the GENEActiv tri-axial accelerometer was worn on the wrist of the non-dominant hand for two 24 hour periods, one during a weekday and another during a weekend.” See lines 232-235.

Line 223: Please add information about “MCS” in brackets

RESPONSE: We have put the relevant sentence in brackets at lines 163-164.

Line 247: or � of

RESPONSE: We have put the word “of” instead of “or”. See line 187.

Lines 248 – 260: I do recommend to represent the information in a table

RESPONSE: We have moved this paragraph to the results section based on Reviewer 1 recommendations. See sub-heading “Characteristics of Study Sample”.

Line 253: Please add information about „GCSE“

RESPONSE: We have added the sentence “(General Certificate of Secondary Education which is equivalent to a USA high school diploma)” after the term GCSE at line 377.

Line 266: The bracket is missing

RESPONSE: We have put the relevant abbreviation in brackets at line 193.

COMMENT: Methods: Why do you only use 2 items of the SDQ scale? The SDQ consist of several scales, each of which forms a score. Why you didn’t use the score (5 items)?

The measurement of the self-regulation is carried out by teachers & parents (7 yr) or only parents (11, 14 yr). Why wasn't that done consistently? Are the results comparable? I recommend to add this part in the limitation and discussion section.

RESPONSE: The cohort study used the SDQ as a behavioural-screening questionnaire of 25 questions. The answers to these questions can be used to produce five sub-scales (each consisting of five items) referring to emotional health (e.g., many fears, easily scared), behavioural problems (e.g., often lies or cheats), hyperactivity/inattention (e.g., restless, overactive, cannot stay still for long), peer-relationship problems (e.g., picked on or bullied by other children) and pro-social behaviour. Each of the sub-scales can be used on their own or combined further. We were not measuring the specific items related to the sub-scales so we needed to select the items that best represented the two constructs. This method was applied in previous research studies (see Edossa et al., 2018 and Pearce et al., 2016), both using the same cohort data set. To confirm this approach Edossa and colleagues as well as our study applied confirmatory factor analyses for each construct at all data points (see table 2), longitudinal invariance testing (see supplementary table 4) and report the reliability of both constructs at lines 387-398. The measurement of self-regulation was assessed by different adults at each time point (teachers and parents at age 7, teachers at age 11 and parents at age 14). Both parents and teachers were used in the younger years as self-report was less of an option (as noted in the limitations section), whereas as the older years self-report was used. We chose to be consistent with the adult assessment across all time points. We agree with your comments about comparability and we have added the following wording “teacher assessment of self-regulation and task based assessments in relation to academic achievement produce comparable results but not parent assessment of self-regulation” to the limitations sections. See lines 599-601.

We have also added “Although most of these studies used parental assessments of self-regulation, recent meta-analytic research confirmed the relationship even when teacher assessments and task based assessments of self-regulation in relation to academic achievement produce comparable results, with both having larger effect sizes than parental assessments” to the discussion section. See lines 523-527.

COMMENT:Methods: Line 291: Please add number of supplement

Supplement 1: Please adapt the items in order to the manuscript (first: emotional regulation, second: beh. regulation)

RESPONSE: We have updated all lines to refer to the supplementary material in the required format within the text of the document. We have changed the order of the table in Supplementary table 1 to show emotional regulation first and behavioural regulation second.

COMMENT: Methods: 

Lines 292/293: Please delete the blank lines

RESPONSE: We have deleted these lines.

Lines 290/291: Please add number of supplement

RESPONSE: We have updated all lines to refer to the supplementary material in the required format within the text of the document. 

Line 307: Please give more details about the accelerometer measurement (see my comment above)

RESPONSE: We have included the following sentences at this line “At age 7 the Actigraph GT1M uni-axial accelerometer was worn on the right hip for seven consecutive days during waking hours; and at age 14 the GENEActiv tri-axial accelerometer was worn on the wrist of the non-dominant hand for two 24 hour periods, one during a weekday and another during a weekend day.” See lines 232-237.

Line 308: Please add more information about “Euclidean Norm Minus One” and please explain what does mg stand for

RESPONSE: We have added the full term for mg as “milligravitational units” and changed the sentence. See lines 237 and next response below.

Line 311: Are the raw accelerometer data comparable with the Cut-Offs (7 & 11 yr)?

RESPONSE: We have added the following wording to the relevant section “Historically, accelerometers cut points were in the form of ‘counts’, however, recently, there has been a move towards measuring PA using raw acceleration techniques, such as Euclidean Norm Minus One instead of counts. Counts are produced by an algorithm whereas raw data is not processed. One study investigated acceleration cut points comparing the Actigraph and GENEActive in different populations (children aged 7-11 years and adults over 18 years) and found that MVPA for adults should be ≥93mg and ≥190mg for children. Guidance for the adolescent age group has not been confirmed. However, one recent study (Rowlands et al., 2018) applied acceleration ≥200mg for MVPA to 1,669 adolescences citing the aforementioned research paper. Presumably, the Millennium Cohort Study selected 100mg as age 14 years is closer regarding development to adulthood than childhood. Furthermore, The GENEActive and Actigraph accelerometers showed strong agreement in a validation study with intra-class correlation >0.95.” See lines 238-251.

Line 313: The items you describe do not measure frequency of PA. Instead the described items measure enjoyment.

RESPONSE: This is noted at line 227-229. 

“The data set measured parameters which include moderate to vigorous physical activity (MVPA) greater than 60 minutes, enjoyment of PA in different settings and duration of type of PA”

We have added the words “of enjoyment” to further clarify this at line 253.

Additionally, frequency as a missing component of the latent variable PA is noted as a weakness of the study at lines 590-591 see below for wording. 

“Second, the data on the frequency of physical activity was not available. The combination of intensity, frequency and duration better represents the construct physical activity as whole and is likely to predict self-regulation and academic achievement for children who are disadvantaged as they have lower participation rates.”

In the age of 14 years the academic achievement was not measured. Please add this fact to the discussion section.

RESPONSE: We have also added “due to data availability” to lines 471 to the discussion section in relation to physical activity and academic achievement.

Line 357: Please add information about the assessment of the SES. When and how was it assessed?

RESPONSE: We have included the following sentence “The data was collected through parental self-report questionnaire, assessed between 0-5 years of age.” at lines 302-303.

Table 1: Please add a reference in the manuscript (see table 1).

Response: We added a sentence in “Measures” section subsection “Socioeconomic status” to refer to table 1 before its presentation.

Results

In my opinion, the results section starts on line 435. The previous information relates to the methods (statistics).

RESPONSE: We have moved the “Data analysis” section to the Methods section. 

Line 430: Please include the number of the addendum.

Line 437: Please include the number of the addendum.

RESPONSE: We have updated all lines to refer to the supplementary material in the required format.

Figure 1: Please give more information about figure 1. Explain the figure in more detail in the manuscript.

RESPONSE: We have added further clarification to the sentence “Figure 1 present the results of the structural model indicating the relationship between latent factors physical activity, emotional regulation and behavioural regulation in the short term at age 7 and 14, in the long term from age 7 to 11 and whether this association is linked to academic achievement. The model was also tested with SES factors which is shown in figure 2.” at lines 450-455.

Line 529: Please try to specify the possible grounds.

RESPONSE: We have added the following paragraph to support our claim. See lines 491-494.

“One study investigated the differences in learning environments between primary and secondary schools identified a deterioration in the quality of teacher-student interaction on factors such as being helpful, friendly and understanding of freedom behaviours (Ferguson & Fraser., 1998).”

I recommend to add strengths of the study to the limitations part

RESPONSE: We have changed the section “Limitation and Future Directions” to include strengths. We have moved “Future Directions” to the “Conclusion” section.

We have included a paragraph on strengths.

“This study has several strengths. First, its use of multiple time points to identify developmental patterns. Second, the exploration of directional processes using SEM as the set of latent variables represented hypothetical constructs using different instruments. Third, consistent with risk and resilience research, this paper adopted an epidemiological approach applying multiple disadvantage factors in relation to cumulative risk. Finally, the combination of constructs in the child’s environment and individual differences, particularly physical activity and self-regulation, allows us to have a deeper understanding of how a combination of factors contribute to developmental trajectories and how this can translate to interventions and improvement in the teaching environment.”

I recommend to structure a short part “future directions and conclusion” as a separate part

RESPONSE: We have moved “Future Directions” to the end of the “Conclusion” section.

There is also theoretical information in the results section. I recommend to limit yourself to the results. 

RESPONSE: We have removed the following “Simulation studies suggest that multiple imputations produce more adequate results instead of list-wise or pairwise deletion” from the results section.

In the discussion, the results of your own study can be linked to the introduction.

Try to shorten and to structure the discussion section in the same order like the introduction section.

RESPONSE: We have moved a section from the Introduction to the Discussion section based on Reviewer 1 recommendations. Specifically, we have moved “Relations between Self-Regulation and Academic Achievement” part of the introduction to the Discussion section under the heading “Physical Activity, Self-Regulation and Academic Achievement.” This has meant that the Discussion sections has increased but at the same time the Introduction section has been significantly reduced. We have used the two main sub-headings in the Discussion section as recommended to link with the Introduction and and the two research questions. This is consistent with the two sub-headings in the Introduction section as recommended in your recommendations on structure of the Introduction.

There is no sample description: I recommend to add a table which describes the sample (age, sex, ….)

RESPONSE: We have moved a paragraph that describes the sample to “Characteristics of Study Sample” to the Results section based on Reviewer 1 recommendations and included the following sentences. 

“A majority of the participants where white (82%), 4.1% were Pakistani, 3.5% were of mixed race, 3.2% were Indian, 1.6% were Bangladeshi, 2% were Black African and 1.3% were Black Caribbean. The gender split was equal amongst boys and girls (50%).” See lines 367-370. 

The “participants” sub-section also includes the gender and ethnicity split at age 7 which is when the physical activity data was first collected. See line 169. 

There is no consort chart: I recommend to add a consort for transparency of the sample composition

RESPONSE: We have added a chart to show the sample composition in the supplementary material. See S6 Figure.

COMMENT: Discussion: Discussion section should include information that cognitive ability only assessed at 5 year olds. PA was not assessed at 11 year olds.

RESPONSE: We have added the sentence “when controlling for prior achievement and cognitive ability at age five” to the discussion section. See line 473. We have added the sentence “Physical activity was not available at age 11 as the cohort study at this age focussed on issues relevant to children entering adolescence such as risky behaviours.” to the “Measures” sub-section “Physical Activity”. See lines 269-271. This is to be consistent with the other “Measures” sub-section “Academic Achievement”. We have also added “due to data availability” to lines 471 to the discussion section in relation to physical activity.

---

## [Decision Letter · Decision Letter 1]

19 Apr 2021

Investigation of the associations between physical activity, self-regulation and educational outcomes in childhood

PONE-D-20-31387R1

Dear Dr. Vasilopoulos,

We’re pleased to inform you that your manuscript has been judged scientifically suitable for publication and will be formally accepted for publication once it meets all outstanding technical requirements.

Kind regards,

Andrea Martinuzzi

Academic Editor

PLOS ONE

Additional Editor Comments (optional):

Reviewers' comments:

Reviewer's Responses to Questions

**Comments to the Author**

1. If the authors have adequately addressed your comments raised in a previous round of review and you feel that this manuscript is now acceptable for publication, you may indicate that here to bypass the “Comments to the Author” section, enter your conflict of interest statement in the “Confidential to Editor” section, and submit your "Accept" recommendation.

Reviewer #1: All comments have been addressed

Reviewer #2: All comments have been addressed

Reviewer #3: All comments have been addressed

2. Is the manuscript technically sound, and do the data support the conclusions?

Reviewer #1: Yes

Reviewer #2: Yes

Reviewer #3: Yes

3. Has the statistical analysis been performed appropriately and rigorously? 

Reviewer #1: I Don't Know

Reviewer #2: Yes

Reviewer #3: I Don't Know

4. Have the authors made all data underlying the findings in their manuscript fully available?

Reviewer #1: Yes

Reviewer #2: Yes

Reviewer #3: Yes

5. Is the manuscript presented in an intelligible fashion and written in standard English?

Reviewer #1: Yes

Reviewer #2: Yes

Reviewer #3: Yes

6. Review Comments to the Author

Reviewer #1: The authors have considered my comments, and this version of the manuscript is acceptable for publication.

Reviewer #2: No comments to authors. It's good piece of work. Look forward to more research work on improving physical activity.

Reviewer #3: Thank you for your detailed revision. You adressed all my comments very well. I appreciate all the work you have done on the manuscript.

7. PLOS authors have the option to publish the peer review history of their article (what does this mean?). If published, this will include your full peer review and any attached files.

Reviewer #1: No

Reviewer #2: No

Reviewer #3: **Yes: **Fabian Kleinke

---

## [Editor Report · Acceptance letter]

21 Apr 2021

PONE-D-20-31387R1 

Investigation of the associations between physical activity, self-regulation and educational outcomes in childhood 

Dear Dr. Vasilopoulos:

I'm pleased to inform you that your manuscript has been deemed suitable for publication in PLOS ONE. Congratulations! Your manuscript is now with our production department. 

Kind regards, 

on behalf of

Dr. Andrea Martinuzzi 

Academic Editor

PLOS ONE